# Research on Land Use Simulation of Incorporating Historical Information into the FLUS Model—Setting Songyuan City as an Example

**Jun Zhang, Zhaoshun Liu and Shujie Li \***

College of Earth Sciences, Jilin University, Changchun 130061, China; jzhang20@mails.jlu.edu.cn (J.Z.);
zhaoshun@jlu.edu.cn (Z.L.)
**\*** Correspondence: lisj@jlu.edu.cn

**Abstract:** Historical information has been included in the study of land use change, but the historical information is described from the perspective of urban growth. This study selected the relevant angle between historical construction land and current construction land, and quantitatively described history information. This research put forward the STLEI index and STEWMEI index, which quantitatively describes the historical information scores, and draw the following conclusions: the closer the construction land generation period is to the current, the greater the driving effect, and this difference is particularly obvious in the past 5 years. We incorporated historical information into the FLUS model; the FoM index increased by 1.93% compared with the previous inclusion, and the Kappa index increased by 2.45% compared with the previous inclusion, verifying the driving role of historical information in land use change. Historical information has an obvious driving effect in land use change. After that, we used the FLUS model, combined with the Markov chain model, anti–planning concepts, and incorporated historical information to simulate the land use change in 2025. These experiments show that historically generated construction land plays a driving role in current and future land use changes and provides a new perspective for the study of land use simulation.

**Keywords:** land use simulation; FLUS model; historical information; Songyuan City

## 1. Introduction

Land provides the wealth of production and living materials, production and living places for human life, and is the foundation of all human life activities. Land resources are non–renewable, and it is difficult to change the type of land use and difficult to reverse. The change of land use type will have a huge impact on economy, society, and ecology [1,2]. In recent years, there have been more and more studies on land use simulation, and many scholars have proposed different models to simulate land use. Mitsova Diana combined urban growth with development zone protection and proposed a new type of cellular automata model [3]. On the basis of analyzing the characteristics of towns and villages land using spatial patterns, Sang Lingling applied the Markov model and CA model to simulate town and village changes [4]. Zhou Fang used CLUE–S and SWAT models to couple the hydrological response of urbanization at different spatial scales in the Yangtze River Delta [5]. Chen Yuming further proposed the Patch–Logistic–CA model on the basis of the Logistic–CA model, which made up for the problem that the Logistic–CA model without considering the spatial evolution of land–use patches causes [6]. Based on GIS and neural networks, Pijanowski simulated a national scale urban growth simulation [7]. Liang Xun considered the impact of urban planning on land use and proposed the CA based future land–use simulation model [8]. Struck Julia simulated and depicted the trajectories of land use change in Europe from 2000 to 2040 [9]. Yang Jie combined geospatial partition, a Markov chain, multi–layer perceptron artificial neural network and cellular automata to extract land type conversion rules and more effectively simulated future urban land use [10].

Liu Xiaoping established a FLUS model based on the original CLUS model, introducing roulette, neighborhood effects and other theories. Compared with the original CLUS model, the simulation accuracy of this model had been further improved [11]. With its high degree of fitting, simple and easy operation, the FLUS model inspired many scholars to study land use simulation [12–16].

Expansion of construction land as the main driving force of land use change [17] received widespread attention from scholars [18–20]. On the basis of summarizing the achievements of predecessors, Li Guangdong put forward five types of driving factors: socio–economic factors, physical factors, neighboring factors, neighborhood factors, land use policies and urban planning to jointly determine the expansion of construction land [21]. Research in recent years found that historical information factors are also important driving factors for changes in land use types; that is, compared with areas developed in earlier years, the surrounding areas of recently developed areas are more likely to be developed in the near future [22–24]. Wang Haijun and Li Xuecao clearly pointed out that historical information is also a driving factor for land use changes [25,26]. However, most of the current research on historical information of land use changes only considered that the historical trend of land use will have an impact on current land use, and no one clearly pointed out that construction land generated in different periods will have different effects on current and future land use changes. This research aims to study the impact of construction land in different periods on current and future land use changes, incorporate it into the FLUS model, explore the impact of historical trend driving factors on the results of land use simulation, and perform future land use simulations, provide a new perspective for the study of land use simulation.

## 2. Materials and Methods

### 2.1. Study Area

Songyuan City is located in the central and western part of Jilin Province (Figure 1), in the triangle of Harbin, Changchun, and Daqing, at the southern end of the Songnen Plain, on the banks of the Songhua River, adjacent to Changchun City and Siping City in the south, and Baicheng City in the west. It is bordered by Tongliao City in Inner Mongolia and faces Heilongjiang Province across the Songhua River to the north. Together with Baotou, Hohhot and Ordos, they are called the "Four Tigers of Economic Growth in Northern China". With the economic growth, Songyuan City's construction land has continued to grow rapidly in recent years. Therefore, this study takes Songyuan City as an example to explore the impact of historical information on land use changes.

### 2.2. Data Source and Processing

The land use data were downloaded from the Geospatial Data Cloud (http://www.gscloud.cn/search, 1 October 2021) from the 2010–2019 Landsat series of remote sensing data, and then obtained by remote sensing interpretation. In addition, Table 1 provides the source of all data obtained in this study.

The selected Landsat series of remote sensing data are all taken in summer, with cloud content ≤5%, and the obtained land use data having a high degree of credibility. The railway data, highway data, central area data, county central area data, ecological protection area data, and permanent basic farmland data provided by Songyuan Natural Resources Bureau are all data for 2020.

The Euclidean distance tool in ArcGIS was used to calculate the distance to the railway, road, water area, city center, and county center. The slope tool in ArcGIS was used to calculate the slope data through DEM data, and standardize them to obtain land use change factors figure (Figure 2).

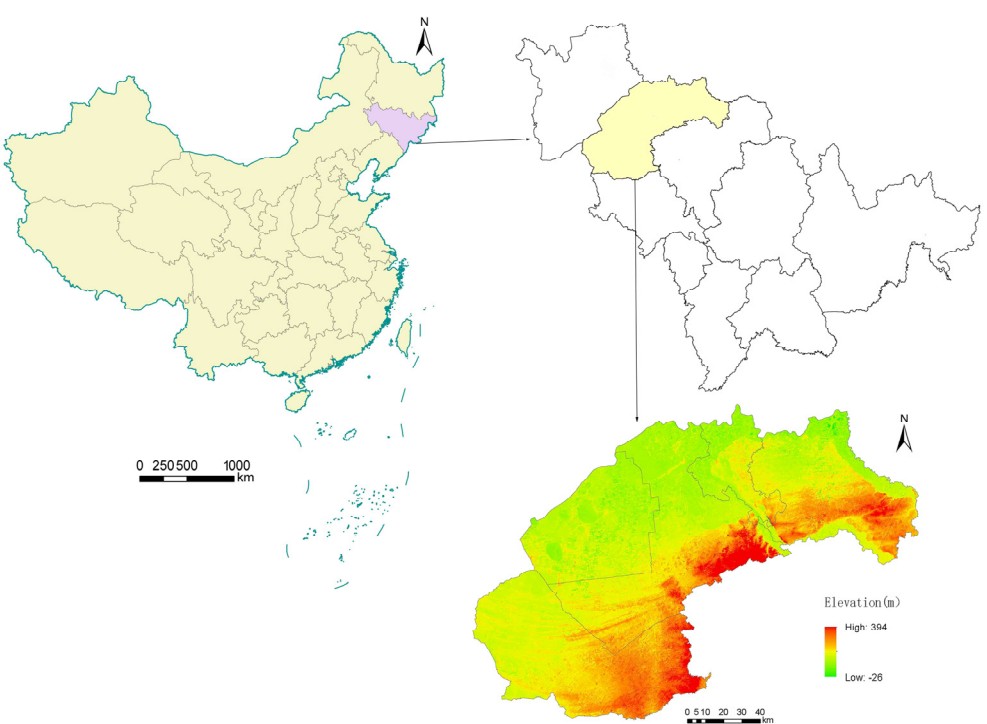

**Figure 1.** The location of Hengyang city.

**Table 1.** Data resources.

| Types | Data Sources |
|---|---|
| Land use data | Geospatial Data Cloud (http://www.gscloud.cn/search, 1 October 2021) |
| DEM data | Geospatial Data Cloud (http://www.gscloud.cn/search, 1 October 2021) |
| Slope data | Geospatial Data Cloud (http://www.gscloud.cn/search, 1 October 2021) |
| Water area data | Geospatial Data Cloud (http://www.gscloud.cn/search, 1 October 2021) |
| Railway data | Natural Resources Bureau of Songyuan City |
| Road data | Natural Resources Bureau of Songyuan City |
| Downtown data | Natural Resources Bureau of Songyuan City |
| County center data | Natural Resources Bureau of Songyuan City |
| Ecological conservation area data | Natural Resources Bureau of Songyuan City |
| Permanent basic farmland data | Natural Resources Bureau of Songyuan City |

*2.3. Methods*

The research method was divided into five steps. The first step is to verify the influence of historical information in the process of expansion of construction land. The second step is to determine the historical information score. The third step is to use the FLUS model to simulate the land use data of Songyuan City in 2019 based on the land use data of Songyuan City in 2016, and compare the historical information and the degree of data fitting that does not consider the historical information. The fourth step is to analyze the specific impact of historical information on the spatial distribution of land use. The fifth step is to use the FLUS model, consider the historical information, and simulate the land use data in 2025 based on the land use data in 2019.

2.3.1. Specific Period Area–Weighted Mean Expansion Index

Liu Xiaoping proposed the Landscape Expansion Index (LEI) [27], which is considered to be a landscape index that can measure the dynamic process of urban growth and has been applied in many cases. Specifically, it is used to construct a buffer zone for newly added

urban patches and intersect with the original urban patches to calculate the proportion of the area of the original cities in the buffer zone. The specific formula is as follows:

$$LEI = 100 \times \frac{A_o}{A_o + A_v},$$ (1)

where LEI is the landscape expansion index of the newly added urban patch, $A_o$ is the intersection of the buffer zone and the occupied category, and $A_v$ is the intersection of the buffer zone and the vacant category.

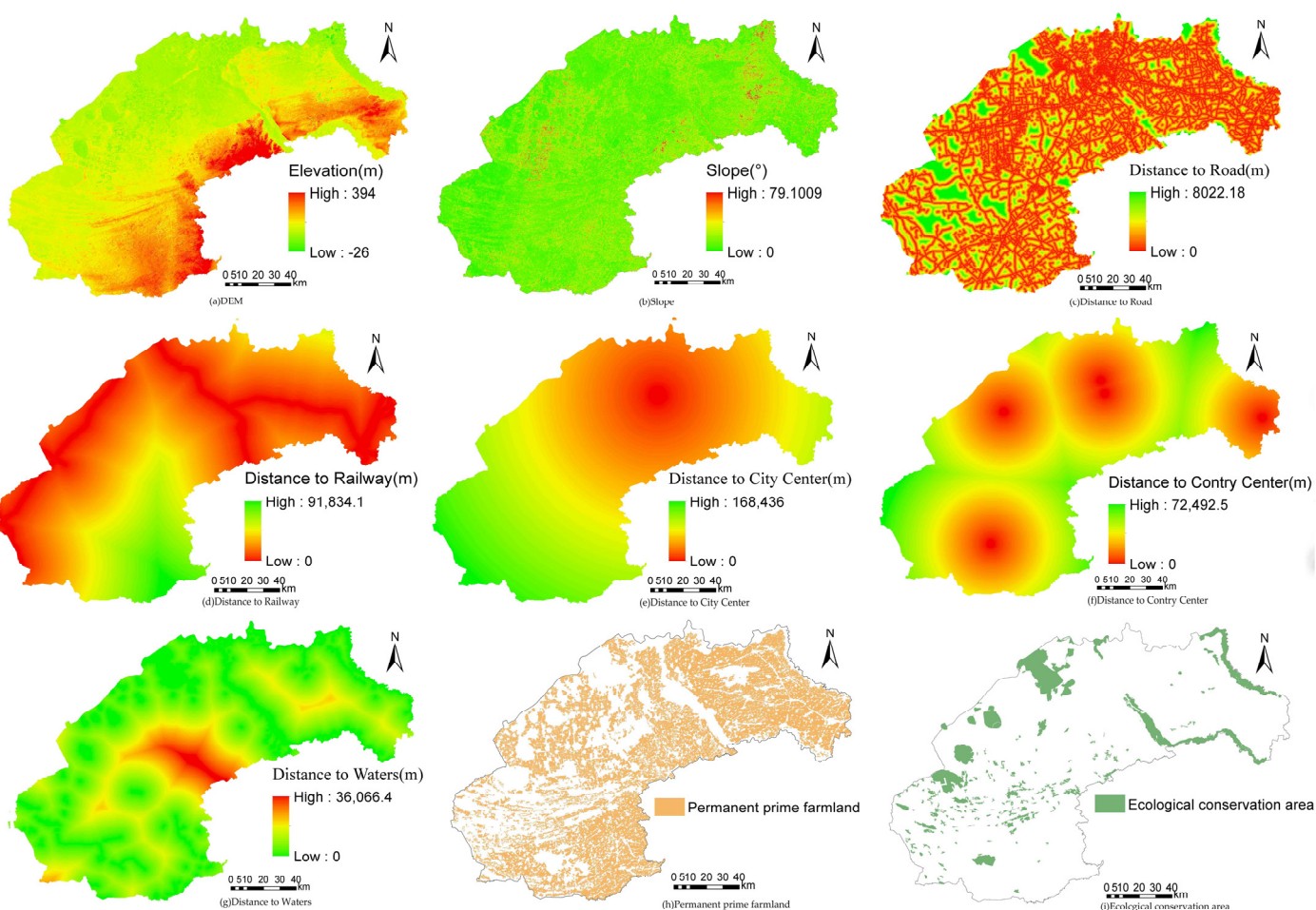

**Figure 2.** Land use change factors.

On the basis of the Landscape Expansion Index (LEI), Liu Xiaoping proposed the Area–weighted Mean Expansion Index (AWMEI) [27]. Specifically, the LEI values of all newly added urban patches are weighted and averaged by area. The index can intuitively reflect the expansion characteristics of the entire city. The specific formula is as follows:

$$AWMEI = \sum_{i}^{N} LEI_i \times \left(\frac{a_i}{A}\right),$$ (2)

where AWMEI is the Area–weighted Mean Expansion Index of the entire city, $LEI_i$ is the LEI of newly added urban patches, $a_i$ is the area of newly added urban patches, and A is the sum of the area of all newly added urban patches.

Based on the LEI and AWMEI, this study proposes the Specific Period Landscape Expansion Index (STLEI) and the Specific Period Area–weighted Mean Expansion Index (STAWMEI). The specific formulas are as follows:

$$STLEI_t = 100 \times \frac{A_t}{A_t + A_v} \tag{3}$$

$$STAWMEI_t = \sum_j^N STLEI_{ij} \times \left(\frac{a_j}{A}\right), \tag{4}$$

where $STLEI_t$ is the Specific Period Landscape Expansion Index of the new construction land patch corresponding of t period, $A_t$ is the intersection of the buffer zone generated by the newly added construction land patch and the construction land patch generated in t period, and $A_v$ is the buffer zone and the vacant category. The $STAWMEI_t$ intersection is the Specific Period Area–weighted Mean Expansion Index of the entire city corresponding of the t period, $STLEI_{ij}$ is the $STLEI_t$ of the newly added construction land patch, $a_j$ is the area of the newly added construction land patch, and A is the sum of the area of all newly added construction land patches.

The STAWMEI value can well reflect the contribution of the construction land patches generated in a certain period to the newly added construction land patches. The higher the STAWMEI value, the higher the contribution of the construction land patches generated in a certain period to the newly added construction land patches.

### 2.3.2. Historical Information Score

The STEWMEI values of newly added construction land patches in different years can reflect well the degree of contribution of construction land patches generated in different years to the newly added construction land patches. Therefore, the historical information score (HIS) can be determined by the STAVMEI value of the newly added construction land patch in different years.

Regression processing was performed on the STEWMEI values of newly added construction land patches in different years. The research tried several types of functions (including linear, exponential, logarithmic, and quadratic functions) through simple regression analysis. Finally, the function with the greatest explanatory power (coefficient of determination, R2) was selected to describe the STSWMEI value in different years.

According to the first law of geography, everything is related to other things, but nearby things are more related to each other [28]. The original construction land patch can greatly promote the generation of adjacent newly added construction land patches. In this study, 1km is considered to be the maximum distance that the original construction land patch has an impact on the newly added construction land patch.

The Historical Information Score (HIS) is specifically as follows:

$$STHIS_t = \begin{cases} (1000 - ED_t) \times RSTAWMEI_t, \ ED_t < 1000 \\ 0, \ ED_t \geq 1000 \end{cases}, \tag{5}$$

$$HIS = Max\{STHIS_1, STHIS_2, STHIS_3, \ldots\}, \tag{6}$$

where $STHIS_t$ is the historical information score of a specific grid unit in a specific period of t time, $ED_t$ is the minimum Euclidean distance (m) from a specific grid unit to the generation of construction land patches at t period, and $RSTAWMEI_t$ is STEWMEI value of the t period after regression analysis, HIS is the historical information score of a specific grid unit.

### 2.3.3. GeoSOS–FLUS Model

The GeoSOS–FLUS model was developed by Liu Xiaoping's team [11] and is suitable for the simulation of future land use changes. The model uses artificial neural networks to train and estimate the probability of each land use type on a specific grid unit, and

a carefully designed adaptive inertia and competition mechanism is used to solve the competition and interaction between different land use types. Through the above two steps, the combined probability of all land use types on each specific grid unit is estimated. In the land use simulation, the model introduces adaptive inertia and competition mechanism, has high simulation accuracy, and can obtain results similar to the actual land use distribution.

2.3.4. Demand Forecast Based on Markov Model

Applying the FLUS model to predict future land use conditions requires future land type quantitative data. Changes in land use are affected by human activities, are complex, and are considered a random process [29]. The Markov model applies Markov analysis to numerically simulate the changing law of random time events in the future. It can predict future land use demand through the land use data of previous years and the current land use data. The following equations can be used specifically, expressed as:

$$X_{(t+1)} = X_{(t)} \times P, \tag{7}$$

## 3. Results

### 3.1. Analysis and Verification of Historical Information Impact

3.1.1. Historical Information Analysis

According to the 2010–2019 land use data of Songyuan City, the generation year of construction land is determined (Figure 3).

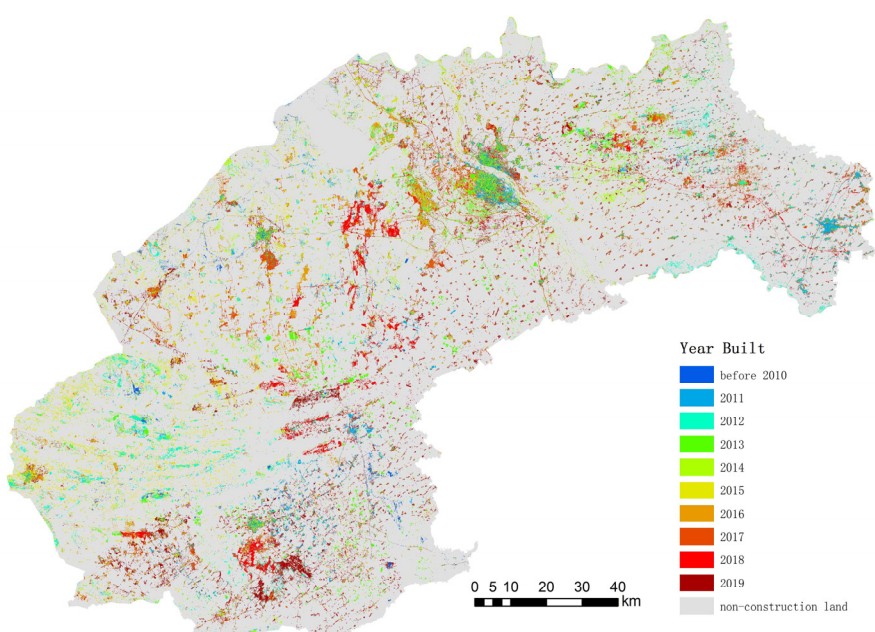

**Figure 3.** Year of completion of construction land in Songyuan City.

The 2012–2018 STEWMEI values of newly added construction land patches in 2019 were 1.79, 2.59, 2.17, 3.20, 4.61, 5.66 and 8.32. It can be seen that the difference in the STEWMEI value of the newly added construction land patch in the last five years is relatively obvious, while the difference in the STEWMEI value of the newly added construction land patch of more than five years is not obvious. It shows that the construction land patches generated in the past five years have increased the driving effect of the newly added construction land patches year by year, and the driving effect of more ancient construction land patches on the newly added construction land patches is not obvious. This proves the existence of historical information, and the historical information driving force of construction land patches generated within five years is particularly significant.

### 3.1.2. FLUS Model Simulation Verification

The land use data of Songyuan City in 2016 was, the FLUS model was used, and under the same premise of other conditions, historical information and no historical information as variables were use, and the land use data of Songyuan City in 2019 were simulated.

By drawing on previous achievements and combining the principle of data availability, in addition to historical information, seven driving factors were selected, including elevation, slope, distance from the city center, distance from the county center, distance from the road, distance from the railway, and distance from the water area. It was obtained by standardization directly based on the map of land use change factors.

The 2012–2016 STAMPMEI value of newly added construction land patches in 2016 was calculated; because the contribution of construction land patches generated more than 5 years after the newly added construction land patches to the newly added construction land patches is not big, it was considered that STAVMEI before 2012 was STAUMEI2012. The ArcGIS software was used to calculate the HIS value of each grid cell in Songyuan City, and standardized processing performed to obtain the 2016 historical information score.

Using the FLUS model, the 2016 land use data were used as the initial data, and the historical and non–historical information in the driving force factor was used as variables to simulate the 2019 land use data (Figure 4).

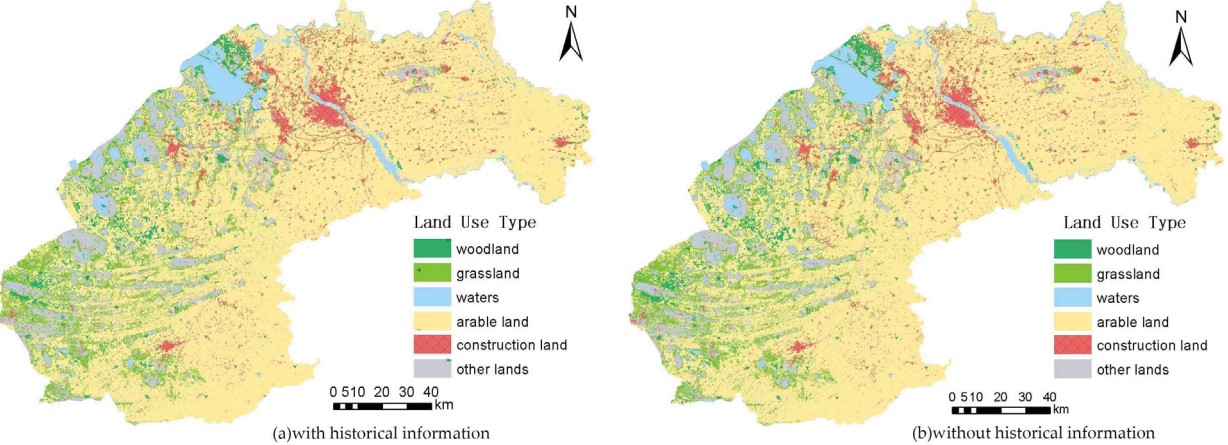

**Figure 4.** Comparison of land use simulation with or without historical information.

The Figure 4a is the result of land use simulation with historical information, and Figure 4b is the result of land use simulation without historical information. Compared with the simulation results of land use without historical information, the FoM index of the simulation results of land use with historical information increased by 1.93%, and the Kappa index increased by 2.45%, which proves that the accuracy of land use simulation has been significantly improved after inputting historical information.

### 3.1.3. Analysis of the Buffer Zone in the Concentrated Built–Up Area

By overlay analyzing the land use simulation situation with historical information and without historical information in 2019, it can be seen that the areas are all construction land, only historical information is simulated as construction land, and only without historical information is simulated as construction land (Figure 5a).

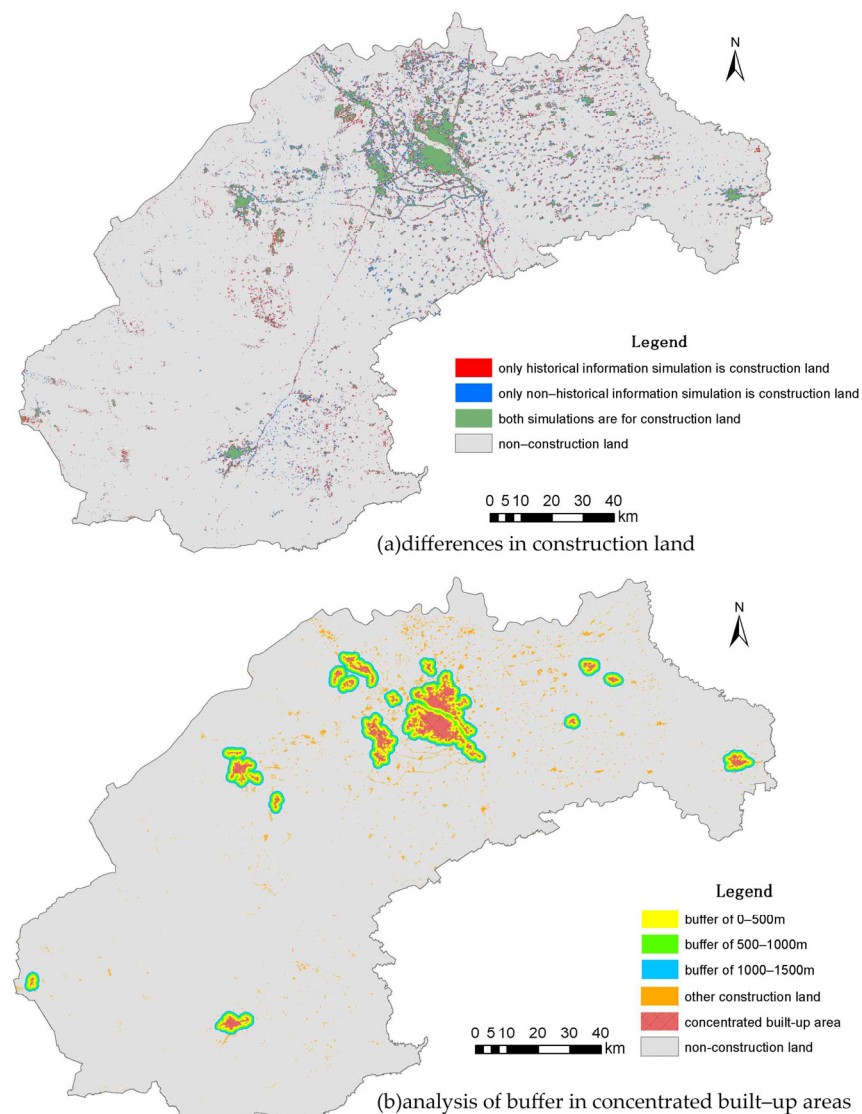

**Figure 5.** Differences in construction land and analysis of buffer in concentrated built–up areas.

In order to better analyze the impact of the presence or absence of historical information on land use simulation, this study introduces the concept of concentrated built–up areas for construction land; the patches with larger area among the patches of construction land in the actual land use situation in 2019 were extracted and used as the concentrated built–up area of construction land (in this study, the patches of more than 2 km² are selected as the larger plates). By performing multi–ring buffer analysis on the concentrated built–up area, buffers of 0–500 m, 500–1000 m, and 1000–1500 m are generated (Figure 5b). The buffer intersected with the area where the construction land differs between the two simulations, and the intersecting area at each distance was calculated separately.

The intersection area of the buffer of 0–500 m, 500–1000 m, 1000–1500 m and the historical simulation construction land is 69.74 km², 18.42 km², 15.34 km², and the intersection area with the simulation construction land without historical information is 59.87 km², 18.55 km², 15.31 km². It can be seen that in the neighborhood of the concentrated built–up area, the amount of construction land generated by historical information simulation is significantly larger than that generated by no historical information simulation, which is especially obvious in the buffer zone of 0–500 m, but in the buffer of 500–1000 m and 1000–1500 m, the amount of construction land generated by the two simulations is not much different. From this, it can be concluded that, after incorporating historical information into

the land use simulation, more construction land will be added around the concentrated built–up area.

### 3.1.4. New Construction Land Density Analysis

According to use of the village as the smallest statistical unit, the density of newly added construction land in the two simulations can be calculated. The formula is as follows:

$$\text{NCD}_i = \frac{\text{NCA}_i}{\text{CA}_i} \tag{8}$$

where $\text{NCD}_i$ is the new construction land density of a village–level administrative unit, $\text{NCA}_i$ is the area of new construction land in this village–level administrative unit, and $\text{CA}_i$ is this village–level administrative unit area.

Figure 6a is the density distribution map of construction land without historical information simulation, and Figure 6b is the density distribution map of construction land with historical information simulation. It can be seen from the figure that the density of construction land with historical information simulation in the central urban area is significantly higher than that of construction land without historical information simulation. From this, it can be concluded that, after adding historical information, the construction land will be further concentrated in the central urban area.

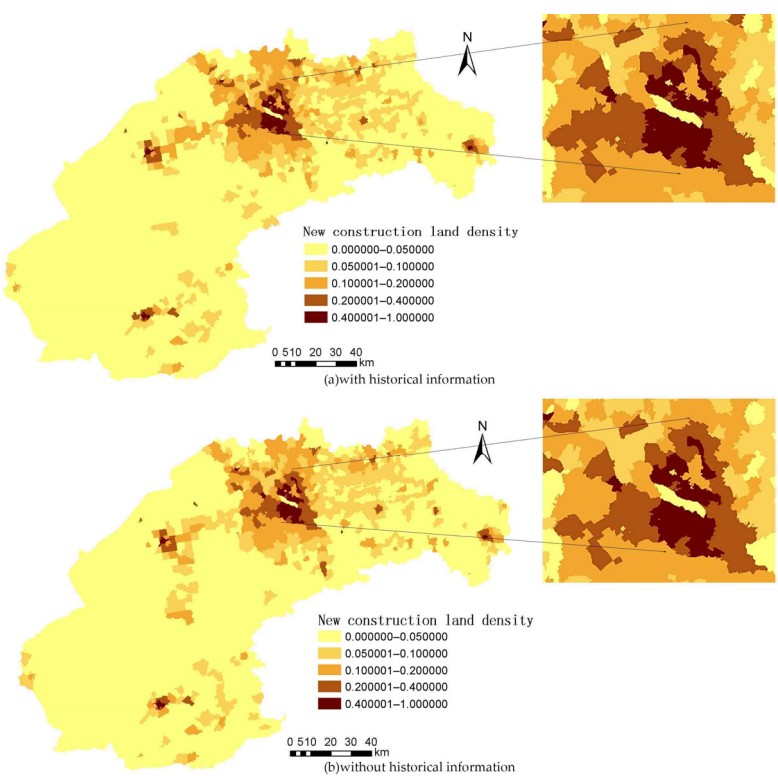

**Figure 6.** Two simulations of construction land density distribution.

### 3.2. Future Land Use Simulation

3.2.1. The Delineation of Non–Construction Area Based on the Idea of "Anti–Planning"

The "anti–planning" concept refers to the delineation of non–built–up areas before urban planning. In this study, ecological protection areas and permanent basic farmland are regarded as non–construction areas, and it is assumed that the land use type will not change in future land use simulations.

According to the permanent basic farmland and ecological conservation area data provided by the Natural Resources Bureau of Songyuan City, the permanent basic farmland protection red line and ecological conservation area red line in Songyuan City are determined.

### 3.2.2. Future Land Use Demand Simulation

Use the Markov chain model, the future land use demand was simulated. The land use data of Songyuan City in 2013 and 2019 were simulated, and the use demand of the land use data of Songyuan City in 2025 were simulated. Usinge the land use data of Songyuan City in 2013 and 2019, a land use transfer matrix (Table 2) was constructed; according to the land use transfer matrix, the Markov chain model was used to determine the land use data of Songyuan City in 2025 (Table 3).

**Table 2.** Land use transfer matrix of Songyuan City from 2013 to 2019.

| 2013 \ 2019 | Arable Land | Woodland | Grassland | Waters | Construction Land | Other Lands | Change Rate |
|---|---|---|---|---|---|---|---|
| Arable land | 11,786.09 | 1277.37 | 962.76 | 114.02 | 800.87 | 332.92 | 22.84% |
| Woodland | 1633.06 | 453.31 | 378.83 | 62.65 | 169.40 | 94.25 | 83.76% |
| Grassland | 430.00 | 222.17 | 381.74 | 81.54 | 89.59 | 22.67 | 68.91% |
| Waters | 111.09 | 56.39 | 59.88 | 641.42 | 4.41 | 10.99 | 27.46% |
| Construction land | 189.27 | 18.73 | 39.68 | 6.56 | 78.21 | 4.17 | 76.77% |

**Table 3.** Forecast of land use demand in Songyuan City in 2025.

| Years | Arable Land | Woodland | Grassland | Waters | Construction Land | Other Lands |
|---|---|---|---|---|---|---|
| 2013 | 154,374.84 | 6891.21 | 29,451.78 | 9195.39 | 3551.56 | 12,798.67 |
| 2019 | 147,781.48 | 5691.51 | 21,240.10 | 9465.67 | 12,111.75 | 19,972.93 |
| 2025 | 146,089.31 | 19,261.68 | 21,961.42 | 10,118.52 | 13,074.13 | 5758.38 |

According to the land use data in 2013 and 2019, using the Markov model, it can be predicted that, by 2025, the area of woodland, waters, grassland and construction land in Songyuan City will increase compared with 2019, and the largest increase is the area of woodland, which is expected to increase by 13,570.17 km$^2$ compared with 2019. The area of other lands and arable land in Songyuan City decreased compared with 2019, and the largest decrease is the other lands, which are expected to decrease by 14,214.55 km$^2$ compared with 2019. It can be seen that, with the further expansion of cities and the implementation of ecological protection policies such as returning farmland to woodland and grassland, the area of woodland, grassland and construction land will be further increased. At the same time, the amount of cultivated land remained basically stable. These reasons will lead to the loss of a large amount of other lands.

### 3.2.3. FLUS Model Simulates Future Land Use

The neural network (ANN) was used to obtain the suitability probability of various land use types within the research scope from the land use data of Songyuan City in 2019 and various driving factors including human activities and natural effects (slope, elevation, distance to road, distance to railway, distance to water, distance to city center, distance to county center, and historical information). The cellular automata based on the adaptive inertia mechanism was used to simulate the land use of Songyuan City in 2025, and the previously set non–construction area was used as the limit data to constrain the change of land use. Combined with previous achievements and the actual situation of land use in Songyuan City, the neighborhood factor parameters of land use type were set as: woodland 0.2, grassland 0.2, waters 0.1, arable land 0.5, other lands 0.1, construction land 1. After parameter setting, the FLUS model to was run obtain the final land use simulation results of Songyuan City in 2025 (Figure 7).

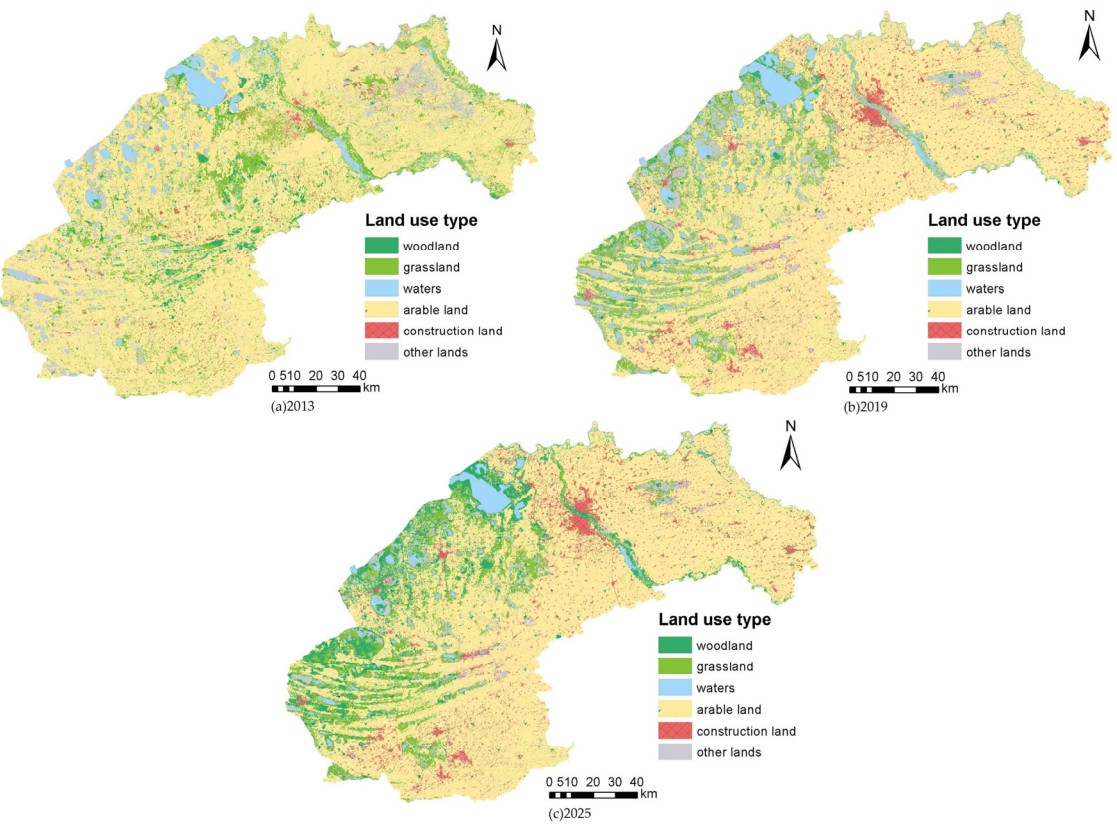

**Figure 7.** Land use change in Songyuan City from 2013 to 2025.

From left to right in Figure 7 are the land use in 2013, the land use in 2019, and the land use simulation in 2025. By comparing the land use data in 2013 and 2019 and the simulated land use data in 2025, we can see that the woodland and grassland in Songyuan City are shifting to the west, and the scale of the forest land was greatly improved. This is consistent with the fact that the main wetland reserves in the study area are concentrated in the west, and the construction of shelter forests has been vigorously promoted in Songyuan City in recent years. In terms of construction land, it not only reflects the trend of outward expansion from the concentrated area of construction land but, at the same time, in the concentrated area of construction land, the density of construction land further increased, which is in line with the current policy requirements for intensive land use. In terms of other lands, the number of other lands in the study area was further reduced, and it was mainly transformed into woodland and grassland.

In general, the simulation results in 2025 continue the basic trend of land use change from 2013 to 2019, and at the same time, meet the basic requirements of "three types of spatial zones and alert–lines" in the current land use master plan, which shows that the simulation results are ideal.

## 4. Discussion

It can be seen from the above analysis that it is completely feasible to describe the historical information in land use from the perspective that the construction land generated in different historical periods will have different impacts on the current land use. In order to quantitatively describe the impact of historical construction land on current land use, this study proposes the STLEI index and the STAWMEI index; these two indices directly reflect the relationship between the generated construction land and the current newly generated construction land in a specific time period, and can intuitively reflect the impact of the construction land generated in different historical periods on the current and future land use. The larger the set time interval, the larger the final value of the STLEI index and the

STAWMEI index will be. Therefore, it is difficult for us to directly compare the magnitude relationship between the STLEI index and the STAWMEI index at different time intervals, and it is also difficult to directly determine the optimal time interval. In order to determine the optimal time interval, we set up a variety of time intervals and conducted multiple experiments. We present two interesting findings; the first is that no matter how the time interval changes, the difference between the STAWMEI indices in two adjacent time periods in the past five years is much greater than the difference between the STAWMEI indices in two adjacent time periods five years ago; the second is that, even when the time interval is set to the minimum (1 year), there is still a large difference between the STAWMEI indices in two adjacent time periods in the past five years. Based on the two findings and the availability of data, we make two assumptions in further research, assumption 1 is that 1 year is the optimal time interval for STLEI index and STAWMEI index; assumption 2 is the same for the STAWMEI index five years ago, that the STAWMEI index five years ago and beyond will remain unchanged.

According to the STAWMEI index in different years and the distance from the construction land generated in the past, the score of historical information is determined. This score can intuitively reflect the driving effect of the construction land generated in the past on the future construction land, and then reflect the driving effect on future land use. Taking the presence or absence of historical information in the driving force factor as a variable, and taking the land use data of Songyuan City in 2016 as the original land use data, using the FLUS model to simulate land use in 2019, it was found that the FoM index and Kappa index of the simulation results with historical information are improved, which proves that historical information can be used as a driving force factor in land use simulation.

## 5. Conclusions

Taking the land use change in Songyuan City from 2010 to 2019 as an example, this study quantitatively describes the historical information through the STLEI index and the STAWMEI index, and further proposes the historical information score. Use the FLUS model to verify the driving effect of historical information on land use status, and the impact of historical information on the spatial distribution of construction land was obtained using the analysis of the buffer for the Concentrated built–up area and the analysis of the density of newly added construction land. Land use data in 2025 were simulated with the FLUS model, demonstrating the complete flow of future land use change simulations after incorporating historical information into drivers. The main conclusions are as follows:

The construction land generated in the past has an obvious driving effect on future land use changes. The closer the construction land generation period is to the present, the greater the driving effect, and the difference is particularly obvious in the past five years. Incorporating historical information into the FLUS model to simulate future land use conditions can significantly improve the accuracy of the simulation and make the simulation more accurate. In terms of the spatial distribution of construction land, land use simulations incorporating historical information will make new construction land more concentrated in central urban areas and other concentrated built–up areas.

In addition, there are still some issues that need to be studied in depth. The expansion of construction land is the main driving force of land use change, but it is not the whole driving force. The change of land use is a complex process, whether there is historical information on the changes of other land types needs further research. Land use is a constantly changing process. During the simulation process, both the currently generated construction land and the future generated construction land will become the construction land generated in the past. Historical information will also change with the generation of construction land. Longer–term land using simulation requires that historical information be no longer fixed, and changes over time to simulate land use changes more accurately. How to continuously adjust historical information in the process of land use simulation still needs further exploration.

**Author Contributions:** Conceptualization, Z.L. and S.L.; methodology, J.Z.; software, J.Z.; validation, J.Z.; formal analysis, Z.L., S.L. and J.Z.; investigation, J.Z.; resources, Z.L. and S.L.; data curation, J.Z. and Z.L.; Writing—original draft, J.Z.; writing—review and editing, J.Z., Z.L. and S.L. visualization, J.Z.; supervision, J.Z.; project administration, J.Z. and S.L.; funding acquisition, S.L. All authors have read and agreed to the published version of the manuscript.

**Funding:** This research was funded by Natural Science Foundation of Jilin Province, China, grant number 20210101395JC.

**Institutional Review Board Statement:** Not applicable.

**Informed Consent Statement:** Not applicable.

**Data Availability Statement:** Not applicable.

**Acknowledgments:** Many thanks to reviewers and the editors for their insightful and constructive comments and suggestions.

**Conflicts of Interest:** The authors declare no conflict of interest.

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
