# Peer review of "Research on Land Use Simulation of Incorporating Historical Information into the FLUS Model—Setting Songyuan City as an Example"

_sustainability, doi:10.3390/su14073828_

Round 1
Reviewer 1 Report
This study aimed at simulating land-use change by incorporating historical information.
For this purpose, the authors have proposed the STLEI index and STEWMEI index to describe the historical information scores and incorporated them into the FLUS model.
However, I think this is the conventional way of land use simulation, the proposed two indices are also based on the transformation of existing studies and lack of innovation.
Besides, from the perspective of the writing of the paper, there are major flaws regarding the literature review, the innovation of this study, and the writing grammar.
In summary, this paper is not suitable for further review and is recommended for rejection.
Author Response
Thank you for your valuable comments.
In this paper, historical information is introduced as the driving factor of land use simulation, which proves the role of historical information in land use and contributes to the simulation of future land use. The STLEI index and STEWMEI index are proposed, and the historical information score is added as a driving factor to the FLUS model, which is the innovation of this paper. he paper has been carefully revised based on the comments of you and other reviewers to improve the quality of the paper.
Thank you again for your valuable comments on this paper, it helps us a lot.

Reviewer 2 Report
I can judge this submitted paper is well organized and the clearly described. Therefore, I will judge this manuscript is almost acceptable in present form.
However, the author(s) can revise the following points according to my comments in order to improve the quality of the paper.
- Legends in the figures: It is quite difficult to read the characters in some figures. The author(s) should rewrite them before submitting final manuscript.
- Figure 1: The author(s) should add a map which can show where Hengyang city locates in China.
- L250-251: “the c concentrated built-up area” What is “c”?
Author Response
Thank you for your valuable comments.
Point 1: Legends in the figures: It is quite difficult to read the characters in some figures. The author(s) should rewrite them before submitting final manuscript.
Response 1: The picture has been remade according to your comments, the font size in the picture has been enlarged, the resolution of the picture has been increased, and the picture has become clearer.
Point 2: Figure 1: The author(s) should add a map which can show where Hengyang city locates in China.
Response 2: We have remade the location map of Songyuan City and added a map of China, which can directly show the location of Songyuan City in China.
Point 3: L250-251: “the c concentrated built-up area” What is “c”?
Response 3: I'm sorry, it is just a typo, we've corrected it. Now, it is in L250-251: “the concentrated built-up area”.
Thank you again for your valuable comments on this paper, it helps us a lot.
Reviewer 3 Report
This study is important and useful to the research community those are working on a very fine scale after including the historical information in the model, though it is a case study over the Songyuan City. Therefore, this paper may be acceptable for scientific publication in the peer-reviewed journal after the following corrections/modifications:
- The introduction needs to improve through language corrections for better understanding to the reader. The last paragraph of the Introduction is considered only one sentence and hence needs to break.
- Figure 1 & 2: Need to improve the color bars values, similarly, other Figure values of the color bar should improve.
- Reference required for calculation of the distance to the railway, road, water area, city center, and county center.
- Line number 100-108 is a single sentence of 9 lines. For more clarity need to break at least 2 to 3 sentences.
- Citation required for equations 2, 3, and 4 used in the study.
- Authors need to provide justification for drastically increasing the Woodland using the model in 2025 compared to the year 2019.
- In Table 3. Need to clarify Moodland or Woodland and in line 296 change km2 to km2
Author Response
Thank you for your valuable comments.
Point 1: The introduction needs to improve through language corrections for better understanding to the reader. The last paragraph of the Introduction is considered only one sentence and hence needs to break.
Response 1: We have reworked the introduction and other sections with language corrections for better understanding to the reader. And based on your comments the last paragraph of the introduction is fused with the penultimate paragraph.
Point 2: Figure 1 & 2: Need to improve the color bars values, similarly, other Figure values of the color bar should improve.
Response 2: Following your comments, we have recreated the pictures in the paper to make them clearer.
Point 3: Reference required for calculation of the distance to the railway, road, water area, city center, and county center.
Response 3: The railway, road, water area, city center, and county center come from Natural Resources Bureau of Songyuan City. We calculated these distance using the distance calculation tool in ArcGIS. The above content has been added to the paper based on your comments.
Point 4: Line number 100-108 is a single sentence of 9 lines. For more clarity need to break at least 2 to 3 sentences.
Response 4: Based on your comments, we've broken this sentence into five sentences so that it doesn't seem too long. now it is in L101-109.
Point 5: Citation required for equations 2, 3, and 4 used in the study.
Response 5: We have added citation to equations 2, but equations 3 and 4 are first proposed in this paper, can we leave out citation?
Point 6: Authors need to provide justification for drastically increasing the Woodland using the model in 2025 compared to the year 2019.
Response 6: We have added to the article the reason why the 2025 simulation results have more forest land than the 2019 results: In recent years, Songyuan City has promulgated a number of policies to plant shelterbelts.
Point 7: In Table 3. Need to clarify Moodland or Woodland and in line 296 change km2 to km2
Response 7: I'm sorry, these are spelling errors in the paper, we rechecked the article and corrected the spelling errors in the article. In Table 3, changed Moodland to Woodland.In L297 change km2 to km2.
Thank you again for your valuable comments on this paper, it helps us a lot.

Round 2
Reviewer 1 Report
I still insist on the view of the first round, because there are still obvious defects in this article after another round of revision. I think it is the author's attitude towards academic paper writing, including:
1) The brief description of the author in the summary part obviously does not meet the norms of academic papers. (eg., Pijanowski, Bryan C. simulated...)
2)Fig 1: It's obviously a Chinese base map.
3)Almost all the pictures can't see the legend clearly.
4)The writing of references is not standardized, and even the picture format such as reference 5 appears.
Anyway, I have expressed my views on the innovation of this paper and the logic of the review in the last round of review, so I won't repeat it here. I hope the author of this paper will uphold academic rigor, seriously treat the writing and publication of papers, and maintain the purity of the academic circle, rather than submitting his own research results in such a perfunctory manner.
Author Response
Thank you for your valuable comments.
Point 1: The brief description of the author in the summary part obviously does not meet the norms of academic papers. (eg., Pijanowski, Bryan C. simulated...)
Response 1: We carefully checked the paper and revised the parts that did not meet the academic standards.
Point 2: Fig 1: It's obviously a Chinese base map.
Response 2: We modified Figure 1 and re-created the map of China, which is no longer used a Chinese base map.
Point 3: Almost all the pictures can't see the legend clearly.
Response 3: We have modified all the pictures, increased the resolution of the pictures, enlarged the size of the pictures, and enlarged the legend. Solved the problem that the legend cannot be seen clearly.
Point 4: The writing of references is not standardized, and even the picture format such as reference 5 appears.
Response 4: We have carefully checked the references section and revised the non-standard parts of the references in strict accordance with the requirements of this journal.
Thank you again for your valuable comments on this paper, it helps us a lot.
